# Cost-Effective Preparation of Hydrophobic and Thermal-Insulating Silica Aerogels

**DOI:** 10.3390/nano14010119

**Published:** 2024-01-03

**Authors:** Jiaqi Shan, Yunpeng Shan, Chang Zou, Ye Hong, Jia Liu, Xingzhong Guo

**Affiliations:** 1ZJU-Hangzhou Global Scientific and Technological Innovation Center, Hangzhou 311200, China; 21626008@zju.edu.cn (J.S.); 21926018@zju.edu.cn (C.Z.); 2State Key Laboratory of Silicon and Advanced Semiconductor Materials, School of Materials Science and Engineering, Zhejiang University, Hangzhou 310027, China; 22126085@zju.edu.cn; 3Zhejiang X-Way Nano Technology Co., Ltd., Hangzhou 311200, China; hy123haoben@outlook.com (Y.H.); liujia-xway@outlook.com (J.L.)

**Keywords:** silica aerogel, hydrophobicity, thermal insulation, water glass, sodium methyl silicate

## Abstract

The aim of this study is to reduce the manufacturing cost of a hydrophobic and heat-insulating silica aerogel and promote its industrial application in the field of thermal insulation. Silica aerogels with hydrophobicity and thermal-insulation capabilities were synthesized by using water-glass as the silicon source and supercritical drying. The effectiveness of acid and alkali catalysis is compared in the formation of the sol. The introduction of sodium methyl silicate for the copolymerization enhances the hydrophobicity of the aerogel. The resultant silica aerogel has high hydrophobicity and a mesoporous structure with a pore volume exceeding 4.0 cm^3^·g^−1^ and a specific surface area exceeding 950 m^2^·g^−1^. The obtained silica aerogel/fiber-glass-mat composite has high thermal insulation, with a thermal conductivity of less than 0.020 W·m^−1^·K^−1^. The cost-effective process is promising for applications in the industrial preparation of silica aerogel thermal-insulating material.

## 1. Introduction

Silica aerogel is a low-density material with a 3D co-continuous nanoporous structure formed by interconnected silica nanoparticles [1]. The co-continuous network formed by silica nanoparticles in aerogels extends heat conduction paths and increases heat radiation loss. The sizes of the nanopores in aerogels are smaller than those in the free path of air molecules, effectively suppressing thermal convection within pores [2,3]. Therefore, silica aerogels often possess outstanding thermal insulation properties. However, due to the low strength of the aerogel itself, it is often necessary to create composite materials with fibers during application [4,5,6,7]. Currently, silica aerogel/glass-fiber composite mats are being utilized as a thermal insulation and fire protection materials in various sectors, including aerospace, petrochemical, construction, and transportation [8,9,10].

In 1930, Kistler synthesized silica aerogels for the first time by utilizing water-glass as the silicon source and incorporating supercritical drying [11]. The study of silica aerogels has spanned over 90 years. During this period, scholars have conducted extensive research focused on the core objective of advancing the industrialization of silica aerogels. They have attempted to utilize tetramethoxysilane and tetraethoxysilane or their oligomers as silicon sources in place of water-glass, achieving success in each endeavor [12,13,14,15,16]. Scholars have also attempted to substitute ethanol with low-surface-tension solvents, such as n-heptane and cyclohexane, and they have adopted less costly drying methods, such as microwave drying and atmospheric drying, to replace supercritical drying [17,18,19,20,21,22,23]. However, the extensive use of alkane solvents has resulted in high recycling and environmental protection costs. Furthermore, the thermal insulation performances of aerogels synthesized by atmospheric drying may decrease. Additionally, the porous structures of silica aerogels ensure that they absorb substantial amounts of water when used outdoors, resulting in significant reductions in their thermal insulation performances. Consequently, aerogels with outstanding thermal insulation performances require hydrophobic modifications. Researchers have treated silica aerogels with solutions of methyl-containing silane coupling agents, such as methyltrimethoxysilane, trimethylchlorosilane, hexamethyldisiloxane, and hexamethyldisilazane, to introduce methyl groups to the surfaces and enhance the aerogels’ hydrophobicity [15,24,25,26,27].

With the in-depth research and industrial development of silica aerogels, a refined industrial process for producing commercial thermal insulation products based on silica aerogels has been established. Currently, global aerogel manufacturers predominantly utilize tetraethoxysilane, known for its stable reaction process, as the silicon source. They then subject it to a series of processing steps, including sols, glass-fiber composites, gels, aging, hydrophobic modifications, solvent replacements, and supercritical drying, to produce silica aerogel composite glass-fiber mats [28,29,30,31]. While the aerogel insulations produced by these processes have secured their positions in the global market, they remain costly, preventing their widespread adoption in price-sensitive sectors such as construction [32,33,34,35]. The production costs of aerogel materials are primarily related to the silicon sources and hydrophobic modifiers. Notably, the latter, with their high consumption levels, low utilization levels, and challenging recycling, pose the greatest obstacles to cost reductions and efficiency enhancements in the aerogel production process [36].

To reduce the cost of raw materials for aerogel synthesis, various silicon sources such as tetramethoxysilane (TMOS), tetraethoxysilane (TEOS), water-glass, and fly ash are used to synthesize silica aerogels. Aerogels synthesized from silicon alkoxide such as TMOS and TEOS exhibit good pore structures and thermal insulation properties, but their prices in the Chinese market typically reach CNY 15–20 per kg, whereas water-glass and fly ash cost only CNY 1–2 per kg and CNY 0.1–0.2 per kg, respectively, in the Chinese market. Given the price differences, water-glass and fly ash offer significant advantages over silanol salts, with fly ash being the most cost-effective. However, fly ash is an industrial waste, and its quality is not controllable. Although there are reports of using fly ash as a silicon source for aerogel synthesis [37,38], the mass production of aerogel products using fly ash remains challenging. Thus, water-glass stands out as one of the few low-cost and quality-controlled silicon sources among the various options. To further reduce the cost of hydrophobic modification, it has been reported that the hydrophobicity of aerogels can be improved by introducing methyl silanol salts such as methyltrimethoxysilane (MTMS) and methyltriethoxysilane (MTES) into the silicon source for copolymerization [39,40,41], without the use of a hydrophobic modifier. Inspired by this, this study explored the possibility of copolymerizing sodium methyl silicate and water-glass to enhance the hydrophobicity of aerogels. Therefore, this study employed cost-effective water-glass and sodium methylsilicate as the silicon sources and hydrochloric acid as the catalyst. By means of the hydrolysis of these two silicon sources and in situ copolymerization, we successfully synthesized hydrophobic silica aerogels without the aid of hydrophobic modifiers, and we also achieved their composites with fiber-glass mats.

## 2. Materials and Methods

Materials: Water-glass (WG, Foshan Zhongfa Water Glass Factory, Foshan, China, 30% solid content and a modulus of 3.37), sodium methyl silicate (SMS, Shanghai yuanye Bio-Technology Co., Ltd., Shanghai, China, 30% solid content), ultra-pure water (H_2_O, Shanghai yuanye Bio-Technology Co., Ltd., Shanghai, China), hydrochloric acid (HCl, Sinopharm Chemical Reagent Co., Ltd., Shanghai, China, 36%~38%, AR), ethanol (EtOH, Sinopharm Chemical Reagent Co., Ltd., Shanghai, China, 99.7%, AR), and glass-fiber mats (GF, Owens Corning Composites (China) Co., Ltd., Hangzhou, China, 10 mm thick, 100 kg/m^3^) were used as obtained.

Preparation process of the silica aerogels: Water-glass (WG) and sodium methyl silicate (SMS) were added sequentially into ultrapure water and stirred for 30 min to produce a uniform precursor solution. After pouring a hydrochloric acid solution (HCl) of a specific concentration into the precursor solution while stirring, a transparent and homogeneous silica sol was obtained by maintaining continuous stirring for 30 min. Afterwards, the silica sol was transferred to a sealed container and placed in an oven at 60 °C, where the gel was aged for 24 h. To replace the solvent of the aged silica wet gel, 10 times the volume of ultrapure water and 2 times the volume of ethanol were used for two additional substitutions. Each substitution was performed at a temperature of 60 °C for a minimum of 12 h. The silica alcohol gel was dried through CO_2_ supercritical drying. The drying process entailed a drying pressure of 17.5 MPa, a drying temperature of 60 °C, and a drying time of 8–10 h. After supercritical drying, the silica alcohol gel was transformed into silica aerogel.

Preparation process of the silica aerogel/glass-fiber composite mat: The silica sol was prepared by mixing WG, SMS, and HCl under stirring, similar to the process used for synthesizing the aerogel. Afterwards, the glass-fiber mat was submerged in the as-prepared silica sol, and it was removed from the sol after 10 min. Then, the removed mat was directly placed into a vacuum bag and kept at 60 °C for 24 h. The solvent replacement and supercritical drying processes for the gel composite glass-fiber mat were identical to those of the silica wet gel, as previously described. After supercritical drying, the silica aerogel/glass-fiber composite mat was obtained.

Characterization of the silica aerogel and silica aerogel/glass-fiber composite mat: The microstructures of the aerogel and the mat were observed by a scanning electron microscope (SEM, Hitachi, Tokyo, Japan, SU8010). The pore structure of the aerogel was measured by a fully automatic surface area and porosity analyzer (BET, Micromeritics, Norcross, GA, USA, ASAP2460). The hydrophobicity capabilities of the aerogel and the mat were analyzed by thermogravimetry (TG, TA, STA6000, temperature range of 30–200 °C and a heating rate of 10 °C/min) and a video-based contact-angle measuring device (DataPhysics, Filderstadt, Germany, OCA20). The thermal insulation performance of the aerogel mat with a thickness of 8 mm was measured by a heat-flow-method thermal conductivity measuring instrument (Netzsch, Selb, Germany, HFM436). The molecular structure of the aerogel powder with a KBr compression was characterized by Fourier-transform infrared spectroscopy (FT-IR, ThermoFisher, Waltham, MA, USA, Nicolet IS5, 400–4000 cm^−1^).

## 3. Results

The hydrolysis polymerization of sodium silicate and sodium methylsilicate can be achieved through acid catalysis or alkali catalysis. This paper reports the synthesis of silica aerogels using two distinct catalytic methods. The synthesis formulas of the aerogel samples are presented in Table 1.

### 3.1. Alkali-Catalyzed Synthesis of the Silica Aerogels

As illustrated in Table 1, the concentrations of HCl for the preparation of samples 1–8 were less than 1.2 mol·L^−1^, and the silica sources employed underwent the sol-gel process under alkali catalysis to form silica aerogels. Figure 1 depicts the microstructures of samples 1–8. As the aging temperature rose (samples 1–3), the microstructures of the synthesized silica aerogels became denser. When the concentrations of HCl decreased (samples 3–6), the silica aerogels exhibited increases in pore size. The macropores with larger pore sizes (more than 1 um) were observable, as shown in the SEM image of sample 6. When an appropriate amount of sodium methylsilicate replaced the water-glass as the silicon source (sample 7), there was no noticeable change in the microstructure of the aerogel. However, when the dosage of sodium methylsilicate was excessive (sample 8), the microstructure of the aerogel exhibited noticeable agglomeration. Figure 2a,b shows the N_2_ adsorption-desorption isotherms and BJH mesopore size distributions of samples 2–5 and 7. Table 2 displays the pore structure characteristics of samples 2–5 and 7. Based on the classification of IUPAC, all N_2_ adsorption-desorption isotherms shown in Figure 2a exhibited type IV, which was characteristic of mesoporous materials. Type IV isotherms exhibit hysteresis loops, and the shapes of these loops in all isotherms were that of type H3, suggesting a slit-like mesoporous structure. As shown in Figure 2b and Table 2, with reductions in the aging temperatures (samples 2, 3), the mesopore diameters and volumes of the aerogels slightly expanded, the mesopore distributions became more concentrated, and the specific surface areas significantly increased. Furthermore, Figure 2b and Table 2 illustrate that as the HCl concentration decreased (samples 3–5), the mesopore diameters, mesopore volumes, and specific surface areas of the aerogels gradually increased. Sample 5 possessed a higher specific surface area and mesopore volume. Using this as a starting point, an appropriate amount of sodium methylsilicate was introduced to the silicon source to prepare sample 7. The specific surface area and mesoporous diameter of sample 7 decreased while the mesoporous volume significantly increased.

### 3.2. Acid-Catalyzed Synthesis of the Silica Aerogels

As shown in Table 1, when the HCl concentrations of samples 9–12 exceeded 1.8 mol·L^−1^, the silica sources used carried out the sol gel process due to acid catalysis, resulting in the formation of the silica aerogels. Figure 3 shows the microstructures of samples 9–12. As the concentrations of HCl increased (samples 9–11), the microstructures of the synthesized silica aerogels exhibited refined frameworks, reduced pore sizes, and enhanced uniformity. Sample 11 possessed a uniform mesoporous structure. Therefore, using the synthesis process of sample 11 as a foundation, an appropriate amount of sodium methylsilicate was introduced into the silicon source to synthesize sample 12. Upon comparing the SEM images of samples 11 and 12, it was evident that the introduction of an appropriate amount of sodium methylsilicate led to increases in the pore sizes of the synthesized silica aerogels, and their microstructures maintained uniform mesoporous structures. Figure 4a,b demonstrates the N_2_ adsorption-desorption isotherms and BJH mesopore size distributions of samples 9–12. Table 3 identifies the pore structures of samples 9–12. All N_2_ adsorption-desorption isotherms shown in Figure 4a were similar to those in Figure 2a, indicating slit-like mesoporous structures. As illustrated in Figure 4b and Table 3, with the increases in the HCl concentrations, the mesopore diameters of the synthesized aerogels slightly decreased while the mesopore volumes and specific surface areas significantly increased. Specifically, the specific surface area and mesopore volume of sample 11 reached 1045.60 m^2^·g^−1^ and 3.10 cm^3^·g^−1^, respectively. After the introduction of an appropriate amount of sodium methylsilicate, the mesopore volume of the synthesized silica aerogel (sample 12) increased to 4.21 cm^3^·g^−1^.

### 3.3. Composition Analysis of the Silica Aerogels

To confirm the incorporation of the methyl groups into the silica aerogels through the addition of sodium methyl silicate, this study conducted thermogravimetry (TG) and infrared spectroscopy (IR) analyses on these materials. Figure 5a presents the TG curves of the silica aerogels. As shown in Figure 5a, samples 7 and 12, synthesized through the introduction of sodium methyl silicate, exhibited lower amounts of weight loss upon being heated to 200 °C. This suggested that these aerogels possessed reduced adsorbed and structural water. This finding indirectly verified the enhancement in the hydrophobicity of these aerogels. Figure 5b shows the IR spectrums of the silica aerogels. The absorption peak near 3430 cm^−1^ corresponded to the antisymmetric stretching vibration of the O–H bond, originating from the Si–OH groups and adsorbed water within the silica aerogels. The absorption peaks near 2965 and 1258 cm^−1^ could be attributed to the antisymmetric stretching and symmetrical bending vibration of the C–H bond, respectively, resulting from the Si–CH_3_. The absorption peak near 1640 cm^−1^ could be ascribed to the bending vibration of the O–H bond from the adsorbed water. The absorption peaks near 1100, 800, and 465 cm^−1^ were due to the antisymmetric stretching, symmetric stretching, and bending vibrations of the Si–O bond, respectively, which originated from the SiO_2_. The absorption peaks near 850 and 758 cm^−1^ corresponded to the planar rocking vibration of the Si–C bond from the Si–CH_3_. It could be seen, as shown in Figure 5b, that the absorption peak corresponding to the O–H bond in the IR spectrum of sample 12, which was synthesized by introducing sodium methyl silicate, was significantly weakened, and the absorption peaks of the Si–C bond and C–H bond from the Si–CH_3_ appeared. This result directly confirmed that the incorporation of methyl groups could be introduced into silica aerogels through the addition of sodium methyl silicate.

### 3.4. Properties of the Silica Aerogel/Glass-Fiber Composite Mats

In order to verify the hydrophobic and thermal-insulating properties of the synthesized silica aerogels, the silica aerogels were compounded with glass-fiber mat, and the thermal conductivity and water contact angles of the silica aerogel composite glass-fiber mats were measured.

Table 4 lists the thermal conductivity capabilities and water contact angles of the various silica aerogel composite fiberglass mats. Sample F0 was a glass-fiber mat without a silica aerogel, serving as a blank control. Sample F5, F7, F11, and F12 were glass-fiber mats composited with the sample 5, 7, 11, and 12 silica aerogels, respectively. As shown in Table 4, the thermal conductivity capability of sample F5 was higher than that of sample F0. The thermal conductivity of sample F7 did not significantly decrease compared to that of sample F0, suggesting that the thermal conductivity ability of sample F7 was lower than that of sample F5. However, its thermal conductivity was not significantly lower than that of air. These results aligned with the previous descriptions of the pore structures of samples 7 and 5. Specifically, after the introduction of sodium methyl silicate, the mesopore volume of the sample 5 silica aerogel synthesized through alkali catalysis increased and its pore structure had improved, yet it still fell short of an ideal silica aerogel structure. Compared to sample F0, samples F11 and F12 exhibited lower levels of thermal conductivity (below 0.020 W·m^−1^·K^−1^). This suggested that both the sample 11 and 12 aerogels had lower levels of thermal conductivity than air. This result aligned with the previous descriptions of their pore structures, indicating that both aerogel samples possessed high specific surface areas and mesopore volumes. The results of the water contact angle tests are illustrated in Figure 6 and Table 4. In comparing samples F5, F11, F7, and F12, it was observed that the silica aerogels synthesized with the addition of sodium methyl silicate exhibited higher contact angles after being compounded with the glass-fiber mats, indicating high levels of hydrophobicity.

## 4. Discussion

In this study, silica aerogels were synthesized through both acid and alkali catalysis, and sodium methyl silicate was incorporated based on an optimized formula. The effects of the sodium methyl silicate dosages, the aging temperatures, and the HCl concentrations in the alkali catalysis system and the HCl concentrations in the acid catalysis system on the microstructures, pore structures, hydrophobicity, and thermal-insulating properties of the silica aerogels and aerogel/glass-fiber composites were investigated in detail.

Firstly, the structures and properties of the aerogels synthesized through alkali catalysis were suboptimal. We speculated that the main reason was that large amounts of NaOH remained in the alkali catalytic systems after the formation of the silica aerogels from the water-glass. This residual NaOH eroded the aerogels during aging. On one hand, this erosion significantly reduced the strengths of the aerogels, resulting in irreversible shrinkage and collapse during the aging and drying processes. On the other hand, it also explained the observation that the aerogels synthesized with lower HCl concentrations in the alkali catalysis systems had larger pore sizes and volumes. Larger mesopore volumes are beneficial for enhancing the thermal-insulating properties of aerogels. However, when the pore sizes were too large, the mesopore volumes in the aerogels decreased, bringing about declines in their thermal-insulating properties. This was because only mesopores smaller than the free path of air can limit air’s thermal convection. Thus, the optimal HCl concentrations in the alkali catalysis systems in this study were established to be 0.9 mol·L^−1^. Furthermore, the aging temperatures significantly impacted the structures of the aerogels in the alkali catalysis systems, which was related to the residual NaOH in the systems. During the aging process, NaOH eroded the aerogels, and we hypothesized that this erosion intensified as the aging temperature rose. Simultaneously, the NaOH erosion may have generated sodium silicate or silica sol particles, which could grow at other locations on the aerogel surfaces during the aging process. This simultaneous erosion and aging process led to a phenomenon akin to “phase separation”, resulting in the microstructures of the aerogels displaying agglomeration. This explained the detrimental effects of the higher aging temperatures on the structures of the aerogel in the alkali catalysis systems. Secondly, the aerogel synthesized with 3.0 mol·L^−1^ HCl in the acid catalysis system exhibited an ideal structure and ideal properties, with its specific surface area and pore volume reaching 1045.60 m^2^·g^−1^ and 3.10 cm^3^·g^−1^, respectively, and the median mesopore diameter was approximately 30 nm. We hypothesized that the primary reason for this was that in the acid catalysis systems, the sodium silicate could undergo sufficient hydrolysis only under low pH conditions, where its rate of hydrolysis outpaced the rate of polycondensation. On one hand, this sufficient hydrolysis ensured the framework strengths of the aerogels, thereby reducing their shrinkage and collapse during the aging and drying processes. On the other hand, the rate of hydrolysis outpacing the rate of polycondensation favored the synthesis of silica aerogels with finer frameworks.

To circumvent the use of costly and inefficient hydrophobic modifiers, this study sought to introduce methyl groups into the silica aerogels through the copolymerization of sodium methyl silicate and water-glass, aiming to achieve their hydrophobic modification. The findings revealed that the addition of an appropriate amount of sodium methyl silicate successfully introduced methyl groups into the silica aerogels, enhancing their hydrophobicity capabilities and increasing their mesopore volumes. However, the excessive addition of sodium methyl silicate caused the agglomeration of the aerogels. We hypothesized that this was primarily due to the strengthening effects of the introduced methyl groups on the overall structures of the aerogels, making them less susceptible to shrinkage and collapse during the aging and drying processes. Given the significant difference in polarity between methylsilicate and water, the introduced methyl groups inherently exhibited hydrophobicity. The excessive introduction of the methyl groups amplified the tendencies for the solid–liquid phase separation during the transitions from water-glass hydrolysis to gelation, resulting in thicker gel skeletons and larger pore sizes. Furthermore, since the methyl groups could not participate in the formation of the Si–O networks in the silica aerogels, this inevitably affected their structural uniformity. Thus, the mass ratio of water-glass to sodium metasilicate of 2:1, which was employed in the synthesis of samples 7 and 12 in this study, was a suitable ratio for synthesis. After the introduction of the methyl groups, the specific surface area of sample 12 remains at 958.58 m^2^·g^−1^, its pore volume increased to 4.21 cm^3^·g^−1^, and its median mesopore diameter was approximately 30 nm.

## 5. Conclusions

In this study, silica aerogels were synthesized by using inexpensive water-glass as the silicon source through both acid and alkali catalysis. The silica aerogels synthesized through acid catalysis exhibited ideal pore structures with specific surface areas of 1045.60 m^2^·g^−1^, mesopore volumes of 3.10 cm^3^·g^−1^, and a median mesopore diameter of approximately 30 nm. To avoid the use of expensive and underutilized hydrophobic modifiers, sodium methyl silicate was incorporated to co-polymerize with the water-glass, and the methyl groups were successfully introduced into the silica aerogels, endowing them with high levels of hydrophobicity. After the hydrophobic modifications, the specific surface areas of the silica aerogels remained at 958.58 m^2^·g^−1^, with mesopore volumes that increased to 4.21 cm^3^·g^−1^ and a median mesopore diameter of approximately 30 nm. The composite of silica aerogels with glass-fiber mats exhibited thermal conductivity capabilities of less than 2.0 W·m^−1^·K^−1^, displaying high thermal-insulating properties.

## Figures and Tables

**Figure 1 nanomaterials-14-00119-f001:**
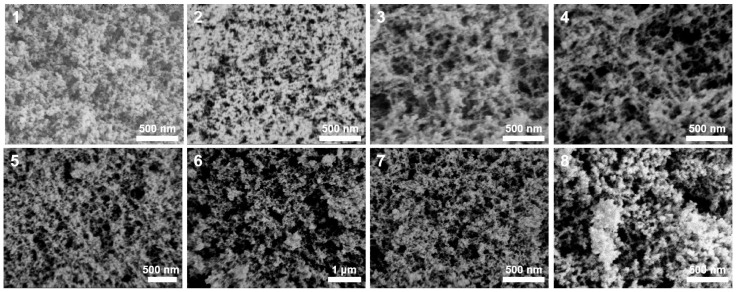
SEM images of the silica aerogels prepared by alkali catalysis.

**Figure 2 nanomaterials-14-00119-f002:**
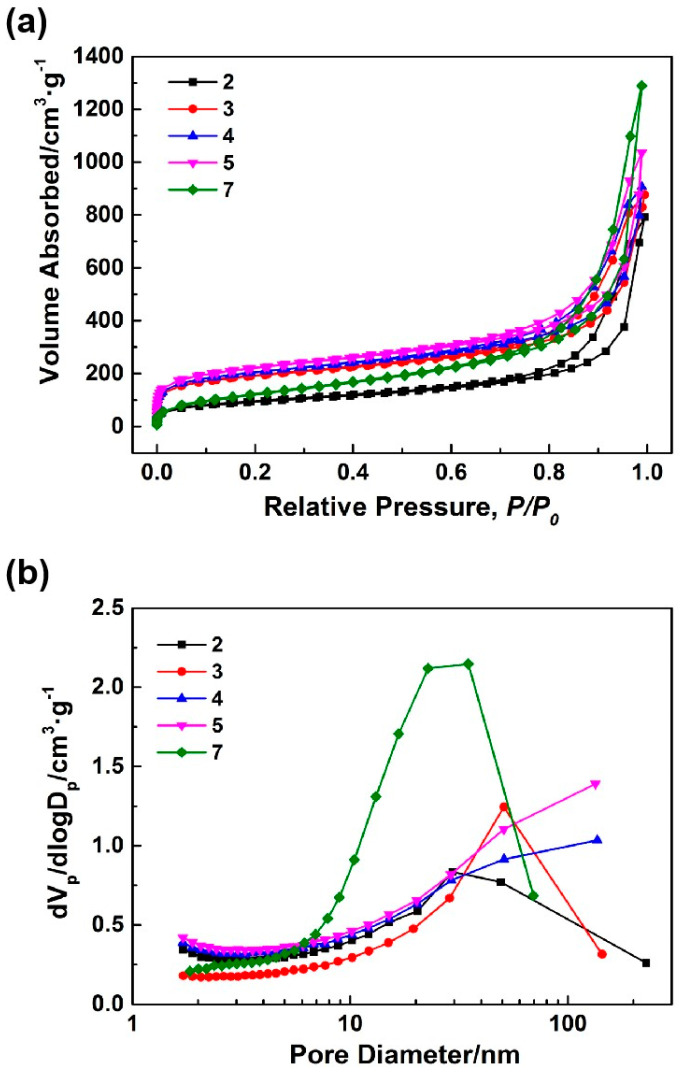
N_2_ adsorption-desorption isotherms (**a**) and BJH mesopore size distributions (**b**) of the silica aerogels prepared by alkali catalysis.

**Figure 3 nanomaterials-14-00119-f003:**
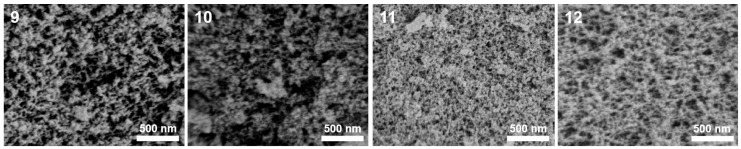
SEM image of the silica aerogels prepared by acid catalysis.

**Figure 4 nanomaterials-14-00119-f004:**
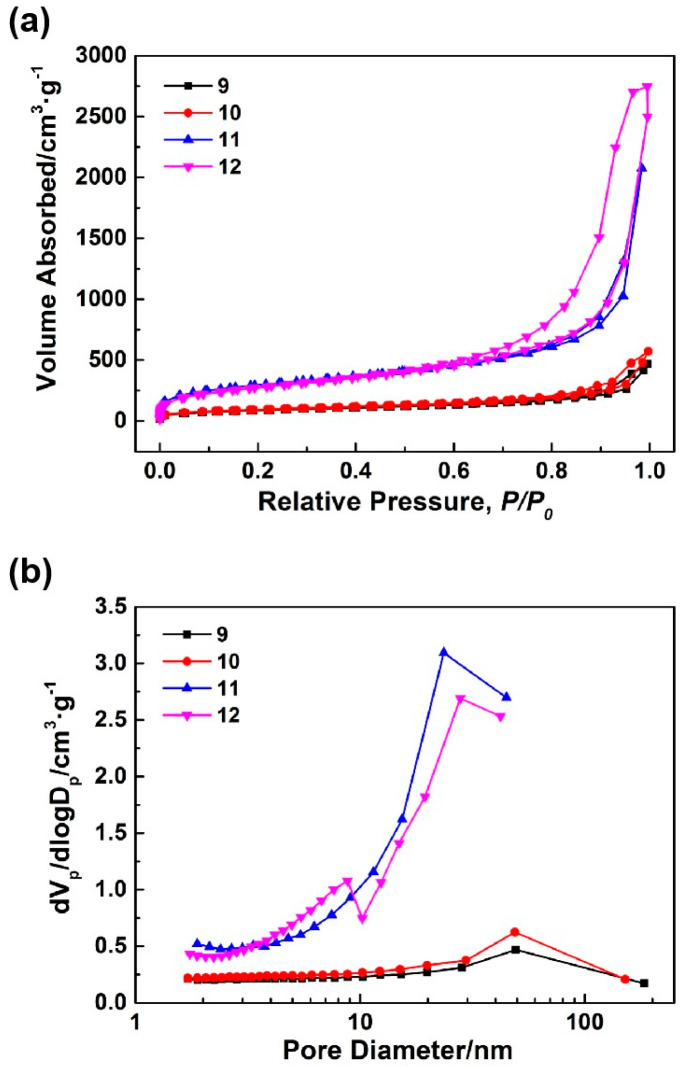
N_2_ adsorption-desorption isotherms (**a**) and BJH mesopore size distributions (**b**) of the silica aerogels prepared by acid catalysis.

**Figure 5 nanomaterials-14-00119-f005:**
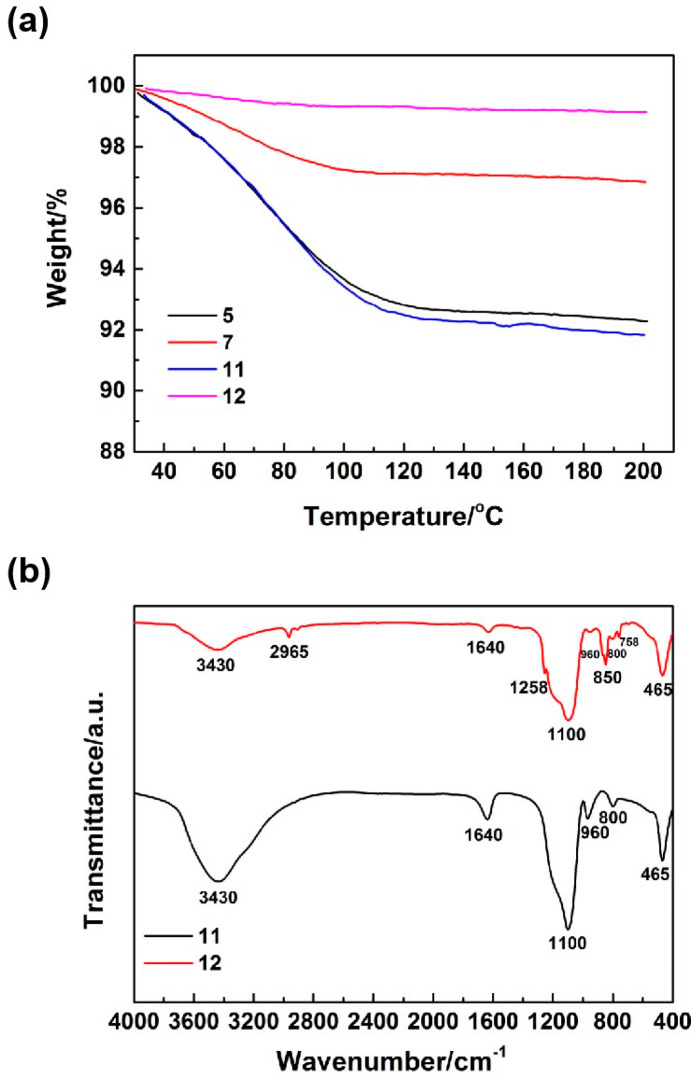
Thermogravimetry (TG) curves (**a**) and infrared (IR) spectrums (**b**) of the silica aerogels.

**Figure 6 nanomaterials-14-00119-f006:**
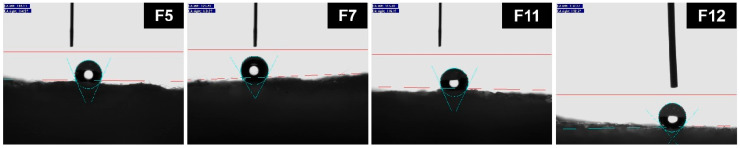
Water contact angle test photos of the silica aerogel/glass-fiber composite mats.

**Table 1 nanomaterials-14-00119-t001:** Synthesis formulas of all the silica aerogel samples.

Catalytic Type	Sample No.	WG/g	SMS/g	H_2_O/g	HCl/g	C_HCl_/mol·L^−1^	T */°C	C_r-NaOH_/mol·L^−1^
Alkali	1	80	0	100	100	1.2	60	0.20
	2	80	0	100	100	1.2	40	0.20
	3	80	0	100	100	1.2	20	0.20
	4	80	0	100	100	1.0	20	0.27
	5	80	0	100	100	0.9	20	0.30
	6	80	0	100	100	0.8	20	0.33
	7	54	26	100	100	0.9	20	0.33
	8	40	40	100	100	0.9	20	0.35
Acid	9	80	0	100	100	1.8	60	0
	10	80	0	100	100	2.4	60	0
	11	80	0	100	100	3.0	60	0
	12	54	26	100	100	3.0	60	0

* T refers to aging temperature and C_r-NaOH_ refers to the concentration of residual NaOH.

**Table 2 nanomaterials-14-00119-t002:** Pore structure characteristics of the silica aerogels prepared by alkali catalysis.

Sample No.	S_p_ ^a^/m^2^·g^−1^	V_pore_ ^b^/cm^3^·g^−1^
2	326.94	1.20
3	654.67	1.22
4	701.46	1.26
5	757.08	1.45
7	466.03	2.00

^a^, Brunauer–Emmett–Teller specific surface area; ^b^, mesopore volume measured by the Barrett–Joyner–Halenda method.

**Table 3 nanomaterials-14-00119-t003:** Pore structure characteristics of the silica aerogels prepared by acid catalysis.

Sample No.	S_p_ ^a^/m^2^·g^−1^	V_pore_ ^b^/cm^3^·g^−1^
9	304.97	0.69
10	331.40	0.85
11	1045.60	3.10
12	958.58	4.21

^a^, Brunauer–Emmett–Teller specific surface area; ^b^, mesopore volume measured by the Barrett–Joyner–Halenda method.

**Table 4 nanomaterials-14-00119-t004:** Thermal conductivity capabilities and water contact angles of the silica aerogel/glass-fiber composite mats.

Sample No.	Thermal Conductivity/W·m^−1^·K^−1^	Water Contact Angle/°
F0	0.031	/
F5	0.037	114
F7	0.029	120
F11	0.020	115
F12	0.018	132

## Data Availability

The data that support the findings of this study are available on reasonable request from the corresponding author.

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
