# Peer review of "Cost-Effective Preparation of Hydrophobic and Thermal-Insulating Silica Aerogels"

_nanomaterials, 2024, doi:10.3390/nano14010119_

Round 1

Reviewer 1 Report

Comments and Suggestions for Authors

The paper ‘Cost-effective preparation of hydrophobic and thermal-insulating silica aerogels’ presents an interesting topic, linked to the reduction of the manufacturing cost of hydrophobic and heat-insulating silica aerogel (i.e. the main weak point of this type of insulation materials with interesting properties). However, the results are rather poor, due to the lack of innovation (previous studies presented similar results), in the the description of the methods, and for lack of standard deviations (possibly omitting part of the results). Minor English errors (grammar or syntax) can also be improved.

Comments:

-    ‘effective water glass and sodium methylsilicate as the silicon sources, hydrochloric acid as the catalyst’: still low sustainable raw materials, which implies economical and environmental costs. Did the author consider alternative, new materials?

-    Methods: details on the conditions, used during the analysis with the different equipments, are completely missing. e.g.: which T scale and heating gradient was used for TGA? which power, conditions or sputtering was used for analysis and specimens? Which method (direct, indirect?) was used for the measurement of the thermal conductivity? Which was the thickness of the specimens when the thermal conductivity was measures? Which part of the spectra (cm-1) was considered for the analyses?

-    SEM images: poor quality, Fig. 1: why 8 images of the same aerogels? Or are those different types of formulations? Please specify in the caption.

-    ‘The specific surface area and mesoporous diameter of Sample 7 decreases, while the mesoporous volume significantly increases’: I understand, however, it is only one sample with no repetition, thus, from my point of view, this is a hint but not a reproducible or reliable result: at least 3 specimens for type of conditions would provide a standard deviation or a clear trend in the data. What about sample 8 in fig. 2b (i.e. its pore size distribution)? Which is the influence of an increasing percentage of sodium methyl silicate in the pore size distribution, in literature?
-    Those two sentences:

‘As illustrated in Table 1, the concentration of HCl for the preparation of Samples 1-8 is less than 1.2 mol·L-1, and the silica source employed undergoes the sol-gel process under alkali catalysis to form silica aerogels. Fig 1 depicts the microstructure of Samples 1-8. As the aging temperature rises (Samples 1-3), the microstructure of the synthesized silica aerogel becomes denser.’

‘As illustrated in Table 1, the HCl concentration for preparing Samples 9-12 exceeds 1.8 mol·L-1, and the silica source employed undergoes the sol-gel process under acid catalysis to form silica aerogels. Fig 3 shows the microstructure of Samples 9-12. ‘

Are basically a copy-paste, please paraphrase in order to increase the soundness and readability.

-    Fig. 5: what is happening to sample 9 and 10 after 50 nm of pore size distribution? Is there another increase which was not shown? Looking at SEM micropictures, it seems so.

-    ‘This result directly confirms that the incorporation of methyl groups can be introduced into silica aerogels through the addition of sodium methyl silicate.’: this was already demonstrated, it should at least be referenced, see e.g.:
-    https://doi.org/10.1016/j.matlet.2018.12.010
-    DOI: 10.1039/c9ra00970a
-    https://doi.org/10.1016/j.powtec.2023.118314

-    Thermals conductivity: again, which was the thickness of the measured specimens? It should be at least 1cm, even if regular insulation board are at least 4cm thick.

Comments on the Quality of English Language

Minor English errors (grammar or syntax) can also be improved.

Reviewer 2 Report

Comments and Suggestions for Authors

Dear authors, 

the manuscript "Cost-effective preparation of hydrophobic and thermal-insulating silica aerogels" submitted to Nanomatterials shows an interesting approach for the preparation of insulating silica aerogels based on glass particles. I have only some minor issues to comment regarding this submission: 

- Authors claim about the cost-effectiveness of the followed approach, but cost analysis or a comparison with other methods are not described. Do you have any approximate idea on the potential cost (and time) savings by the production of these aerogels compared to other processes/materials?

- Have you performed any statistical analysis on the results obtained? It might be useful to know the number of replicas performed per assay and also to show the significance of results. 

- What materials is your proposed intented to substitute? Is there any commercial materials or any other in the literature that would provide similar properties to yours? I think the manuscript needs of some more discussion. 

I hope the comments are useful to improve the manuscript. Best regards

Reviewer 3 Report

Comments and Suggestions for Authors

Thank you for sending the article entitled “Cost-effective preparation of hydrophobic and thermal-insulating silica aerogels”. The authors in this work used inexpensive water glass to prepare silica aerogels with large specific areas and pore volumes a cost-effective process. The resultant aerogels possess high thermal insulation.

There are some points that need to be taken into consideration before publication.

1-      The authors need to mention more precisely the initial concentrations of NaOH and the residual amount that they claim can erode the aerogels. This can be added to Table 1.

2-      In the literature, many reports on silica aerogels demonstrate cost-effective approaches (using fly ash, etc., or different conditions), the authors summarize the previous reports and clarify the advantages of the current research.

3-      SEM photos in Figures 1 and 3 need to improve by increasing the resolution (magnification conditions and sample preparations can be added in the methods section).

4-      Some typographic mistakes need to be revised carefully.

For example, In Table 3 needs to use acid catalysis, not alkali 

Round 2

Reviewer 1 Report

Comments and Suggestions for Authors

Previously provided suggestions to improve the quality of the paper were only partially considered, with minor integration in the manuscript. Please reconsider and improve  the text accordingly

Comments on the Quality of English Language

 Minor editing of English language required

Author Response

Thank you again for your valuable comments on the paper. We have responded to your previous comments one by one and revised the paper based on some of them.  Thank you for your recognition of our previous revision work. We will carefully consider your comments and continue to improve this paper.

Reviewer 2 Report

Comments and Suggestions for Authors

Dear authors,

thanks for assessing the comments from the previous round of revisions. At the moment I have nothing to add. 

Regards,

Author Response

Thank you again for your valuable comments on this paper.